# Sepsis awareness and knowledge amongst nurses, physicians and paramedics of a tertiary care center in Switzerland: A survey-based cross-sectional study

Jean Regina[1], Marie-Annick Le Pogam[2], Tapio Niemi[2], Rachid Akrour[3], Santino Pepe[4], Isabelle Lehn[5], Jean-Blaise Wasserfallen[4], Thierry Calandra[1,6,7], Sylvain Meylan[1]*

1 Department of Medicine, Infectious Diseases Service, Lausanne University Hospital, University of Lausanne, Lausanne, Switzerland, 2 Centre for Primary Care and Public Health (Unisanté), University of Lausanne, Lausanne, Switzerland, 3 Service of Geriatrics and Geriatric Rehabilitation, Lausanne University Hospital, Lausanne, Switzerland, 4 Medical Directorate, Lausanne University Hospital, Lausanne, Switzerland, 5 Director of Nursing, Lausanne University Hospital, Lausanne, Switzerland, 6 Department of Medicine and Department of Laboratory Medicine and Pathology, Service of Immunology and Allergy, Center for Human Immunology Lausanne, Lausanne University Hospital (CHUV), University of Lausanne, Lausanne, Switzerland, 7 Department of Infectious Diseases and Tropical Medicine, Necker-Enfants Malades University Hospital, University of Paris Cité, Paris, France

* sylvain.meylan@chuv.ch

**Data Availability Statement:** The data is available here: https://zenodo.org/record/7031181?token=eyJhbGciOiJIUzUxMiIsImV4cCI6MTY2NDQwMj

## Abstract

### Background

Sepsis is a leading cause of morbidity and mortality. Prompt recognition and management are critical to improve outcomes.

### Methods

We conducted a survey among nurses and physicians of all adult departments of the Lausanne University Hospital (LUH) and paramedics transporting patients to our hospital. Measured outcomes included professionals' demographics (age, profession, seniority, unit of activity), quantification of prior sepsis education, self-evaluation, and knowledge of sepsis epidemiology, definition, recognition, and management. Correlation between surveyed personnel and sepsis perceptions and knowledge were assessed with univariable and multivariable logistic regression models.

### Results

Between January and October 2020, we contacted 1'216 of the 4'417 professionals (27.5%) of the LUH, of whom 1'116 (91.8%) completed the survey, including 619 of 2'463 (25.1%) nurses, 348 of 1'664 (20.9%) physicians and 149 of 290 (51.4%) paramedics. While 98.5% of the participants were familiar with the word "sepsis" (97.4% of nurses, 100% of physicians and 99.3% of paramedics), only 13% of them (physicians: 28.4%, nurses: 5.9%, paramedics: 6.8%) correctly identified the Sepsis-3 consensus definition. Similarly, only 48% and 49.3% of the physicians and 10.1% an 11.9% of the nurses knew that SOFA was a sepsis

M5OSwiaWF0IjoxNjYxNzY4NzM3fQ.eyJkYXRhIj
p7InJlY2lkIjo3MDMxMTgxfSwiaWQiOjl1NjUyLC
JybmQiOil4NTYyZDliNSJ9.vVrZo0V7c30i0yTKjz
DYxr3NtCo4ZKkBwL9SRkGOAwQnHsOzM2x
Hz3IH5zOqBdJKEUhiWSeSIZdCgNtvwdt0kA.

**Funding:** The study was supported by the Société Académique Vaudoise for material acquisition. The funders had no role in study design, data collection and analysis, decision to publish, or preparation of the manuscript.

**Competing interests:** The authors have declared that no competing interests exist.

**Abbreviations:** CI, Confidence interval; ED, Emergency department; HCP, Healthcare professionals; ICU, Intensive care unit; LUH, Lausanne University Hospital; NEWS, National early warning score; OR, Odds ratio; qSOFA, quick sepsis-related organ failure assessment; SIRS, Systemic Inflammatory Response Syndrome; SOFA, Sepsis-related organ failure assessment; SSC, Surviving Sepsis Campaign.

defining score and that the qSOFA score was a predictor of increased mortality, respectively. Furthermore, 15.8% of the physicians and 1.0% of the nurses knew the three components of the qSOFA score. For patients with suspected sepsis, 96.1%, 91.6% and 75.8% of physicians respectively chose blood cultures, broad-spectrum antibiotics and fluid resuscitation as therapeutic interventions to be initiated within 1 (76.4%) to 3 (18.2%) hours. For nurses and physicians, recent training correlated with knowledge of SOFA score (ORs [95% CI]: 3.956 [2.018–7.752] and 2.617 [1.527–4.485]) and qSOFA (ORs [95%CI]: 5.804 [2.653–9.742] and 2.291 [1.342–3.910]) scores purposes. Furthermore, recent training also correlated with adequate sepsis definition (ORs [95%CI]: 1.839 [1.026–3.295]) and the components of qSOFA (ORs [95%CI]: 2.388 [1.110–5.136]) in physicians.

## Conclusions

This sepsis survey conducted among physicians, nurses and paramedics of a tertiary Swiss medical center identified a deficit of sepsis awareness and knowledge reflecting a lack of sepsis-specific continuing education requiring immediate corrective measures.

## Introduction

Sepsis is a syndrome defined as a dysregulation of the host's response to an infection [1]. Its incidence has increased over the past decades and, in 2017, accounted for an estimated 48.9 million cases and 11 million deaths globally, more deadly than stroke and myocardial infarction combined [2]. Sepsis is also associated with significant long-term morbidity, including cognitive impairment, recurrent septic episodes, and increased mortality amongst survivors [3, 4]. In the absence of specific targeted therapy blunting the dysregulated host response to infection, optimal sepsis management relies on rapid recognition, initiation of antimicrobial therapy, and intensive supportive care. Since 2002, the Surviving Sepsis Campaign (SSC) has aimed to reduce sepsis-related mortality and morbidity by increasing sepsis awareness among professionals and providing consensus management guidelines structured into bundles [5–8].

Sepsis awareness and prompt recognition by healthcare professionals (HCPs) are critical components of the management of septic patients. Sepsis awareness includes basic notion of epidemiology, definition of sepsis, and familiarity with the implementation of bedside scoring tools [9]. In the last three decades, sepsis definitions have been reviewed twice since the initial round of 1991 with the last iteration being the 2016 Sepsis-3 consensus definitions [1]. These changes in definitions have been accompanied by changes in the clinical score and diagnostic criteria. As an example, Systemic Inflammatory Response Syndrome (SIRS) is now replaced by the sequential [Sepsis-related] organ failure assessment (SOFA) score. Despite being introduced more than six years ago, there is a dearth of article on the degree of actual knowledge about the actual content of the definition among various HCPs. We identified only three studies on sepsis awareness amongst HCPs. However, the size and scope of HCPs tested on their knowledge of Sepsis-3 consensus definitions were limited [10–12]. Studies of previous sepsis definitions have revealed gaps in sepsis recognition and management amongst medical, nursing and paramedical staff [9, 11, 13–20]. Most studies, however, focus on a single HCP subset, have limited participation (50–200 participants), and are restricted to a single department. Furthermore, few studies have been conducted in wards despite nosocomial sepsis representing 20–30% of all cases [21–23]. We aimed to have a representative understanding of sepsis awareness and knowledge for our tertiary center.

## Material and methods

### Study aim, design and setting

In 2019, the Lausanne University Hospital (LUH) launched a quality-of-care program to improve sepsis management that was part of the 2019–2023 Strategic Plan of LUH. This study, which is a part of this program, aims to quantify Sepsis-3 consensus awareness amongst nurses and physicians of various clinical units at LUH and paramedics transporting patients to our hospital and identify potential deficits that should be addressed in continuing education.

This cross-sectional study was conducted through an anonymous, on-line survey measuring the awareness and knowledge about sepsis among nurses and physicians of the LUH and paramedics transporting patients to our hospital. The LUH is a 1'568-bed tertiary care university hospital, serving the city of Lausanne (population circa 300'000 inhabitants) and the tertiary care reference medical center for the Canton de Vaud (799'145 inhabitants) in Switzerland. At the time of the survey, no department had an active education or clinical practice sepsis sepsis program.

### Measures

The research team designed a survey drawing from previously published surveys assessing knowledge and awareness of sepsis [9, 14, 24, 25]. The questions were tailored to the profession (clinical scenarios adapted to the activity sector—medicine, surgery, emergency department or gynecology). The survey was written and completed in French. Each section of the survey (paramedics', nurses' and physicians' section) was submitted to three focus groups consisting of 3 to 6 participants of all seniority levels of each profession (IE nurses, physicians, and paramedics), commonly involved in care of patients with sepsis. These focus groups assessed the applicability, appropriateness (validity) of the survey and whether formulations and relevance of questions were adequate. The survey was revised using feedback from the groups. Surveys of nursing staff and paramedics were more focused on screening, initial evaluation, and early management whereas physicians were also tested on diagnosis and management. Response options included Likert-type scales, binary (e.g., "yes/no") or multiple choice. Each question was locked upon answering, which prevented post hoc changes that could be influenced by information provided at later stages of the survey. The final survey contained questions on participants' demographic characteristics (5/7/6 questions for nurses/paramedics/physicians), awareness was characterized by questions on sepsis continuing education (3/3/3 questions) and self-evaluation of sepsis knowledge and clinical management (2/2/2 questions); the participants' knowledge was characterized by questions on definitions, scores, and epidemiology (11/12/14 questions), and sepsis management (4/4/5 questions). The survey was developed in REDCap (Research Electronic Data Capture) software so as to automatically export participants' responses to a database [26, 27]. Surveys both in French and in English are provided as supplementary material (supp. meth. Survey S1–S3 Files).

### Data collection and recruitment

Participants were recruited between January 20 and October 10, 2020. We aimed for a convenience sample size of 1'000 persons (approx. 20% of the active HCPs), including registered nurses, and physicians, including medical residents and fellows having graduated from medical school and who were in training for board certification in a medical specialty and attendings, issued from all departments (Emergency department (ED), intensive care unit (ICU), Medicine, Paramedic, Psychiatry, and Surgery) and professions (paramedics, nurses and physicians) in order to achieve maximum representativity of LUH staff considered as HCPs.

Pediatrics and neonatology staff (not covered by Sepsis-3 consensus definitions) as well as nurses and physicians not in daily contact with patients (i.e., those working in research teams or in administration) were excluded. For paramedics, we included those transporting patients to LUH. Undergraduate trainees were excluded. We favored a supervised approach rather than a dissemination of the survey to all HCPs by email. Participants answered the online survey under trained interviewer supervision so as to maximize data quality and to avoid biased responses (internet queries, discussions between colleagues). Furthermore, to avoid multiple answers by an individual HCP, surveys were accessed by QR-code only available at screening; timing of survey completion and email addresses were registered.

Thus, participants were screened amongst the medical (n = 1'664) and nursing staff (n = 2'463) in daily contact with patients of LUH and amongst paramedics transporting patients to our hospital (n = 290) during the screening period. The response rate was defined as the fraction of responders amongst HCPs screened. Screening by trained interviewers took place during scheduled patient hand-offs, seminars, or group meetings, as permitted by heads of units. Participation was voluntary and anonymous. Participants completed the online survey using tablets or smartphones (participants' or provided by the investigators).

## Statistical analysis

We described participant characteristics and survey responses across professions: 1) nurses, 2) physicians, and 3) paramedic. Continuous variables were summarized as means and standard deviations [SD] and categorical variables as frequencies and proportions. We also evaluated study participants representativeness of the LUH population of nurses and physicians using Student t-test and Pearson $\chi^2$ test for comparing mean ages and proportions of female professionals. In order to assess associations between sepsis awareness and proxies of prior medical and sepsis training, we used univariable and multivariable logistic regression models with age (continuous variable), continuing education (yes vs. no or last training < 3 years), professional experience (> 5 years vs. ≤ 5 years), self-evaluation of sepsis knowledge and skills (good-very or good vs. others) and field of practice (ED, ICU, Medicine, Paramedic, Psychiatry, or Surgery) as explanatory variables. For each model, we estimated the odds ratio (OR) of correct vs. incorrect answer as well as is 95% confidence interval (95%CI). All tests for statistical significance were two tailed (p<0.05). We performed statistical analyses using the computing environment R version 4.0.2 (R Development Core Team, 2005) and Prism version 9.0.0 (Graphpad Software).

## Ethics approval

The local institutional review board, the Commission d'Ethique sur la Recherche du Canton de Vaud (CER-VD, Lausanne, Switzerland) viewed the project as a quality of care and did not require written consent for this research project (Decision REQ-2019-01072 on 28.10.2019).

## Results

### Participants

Among the 4'417 eligible health-care professionals (HCPs) comprising 290 paramedics, 2'463 nurses and 1'664 physicians, 1'216 (27.5%) were contacted for participation, of whom 1'116 (91.8%) completed the survey (46 refused to participate and 54 were excluded because of incomplete answer) (Fig 1A). All clinical departments were included, though representation of profession and specialty varied within the departments (Fig 1B). Table 1 shows the characteristics of the participants. The mean age of participating nurses was not different from the

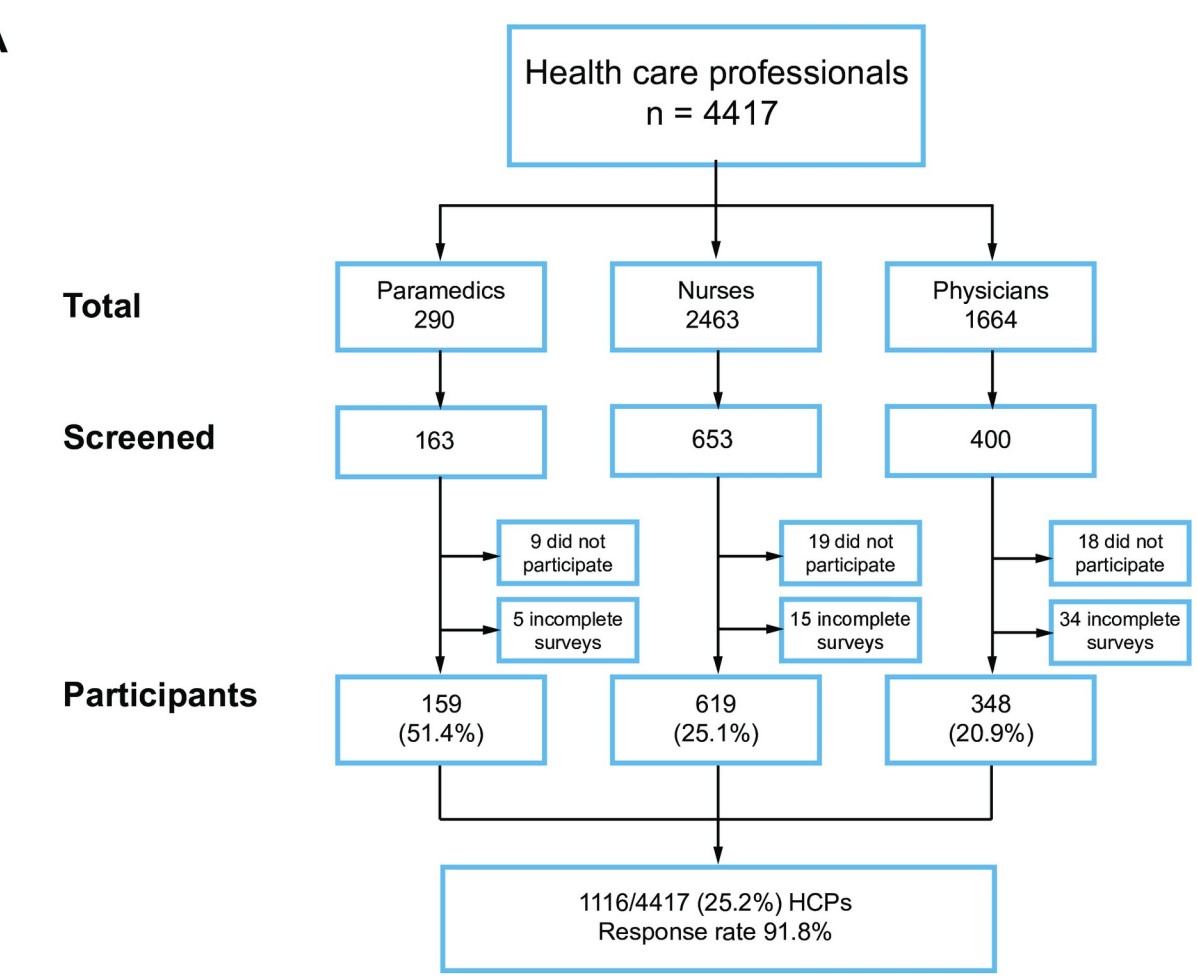

**B    HCPs distribution in the tertiary center**

|  | Nurses | Physicians |
|---|---|---|
| **ED** | 73/132 (55.3%) | 30/91 (33.0%) |
| **ICU** | 73/210 (34.8%) | 23/43 (53.5%) |
| **MEDICINE** | 234/756 (30.9%) | 179/612 (29.2%) |
| **SURGERY** | 208/808 (25.7%) | 98/332 (29.5%) |
| **PSYCHIATRY** | 31/236 (13.1%) | 18/418 (4.3%) |
| **TOTAL** | 619/2142 (28.9%) | 348/1260 (34.8%) |

**Fig 1. Study population.** Flowchart (A) and HCP distribution according to specialty (ED: Emergency Department; ICU: Intensive Care Unit). Surgery encompasses visceral, thoracic, and vascular surgery, neurosurgery, Gynecology, ENT, and orthopedics. RR response rate of participants screened.

**Table 1. Participant's demographical characteristics.**

| | Overall | Nurses | Physicians | Paramedics |
|---|---|---|---|---|
| $n_{participants}$ | 1116 | 619 | 348 | 149 |
| Age (mean [SD]) in years participants/institution | | 38.0 (10.2)/39.1 (10.3) | 35.2 (7.6)/36.4 (9.4) | 41 (9.5)/NA |
| T-test comparison (sample vs. overall) | | *p = 0.01* | *p<0.01* | |
| Female gender (%) sample/ institution | | 75.8/79.7 | 46.8/51.7 | 27.5/NA |
| Distribution difference ($c^2$) (sample vs. overall) | | *p = 0.03* | *p = 0.1* | |
| Medical experience (%) | | | | |
| $<$ 1 year | 79.7 | 43 (6.9) | 29 (8.3) | 2 (1.3) |
| 1–3 years | 117 (10.5) | 58 (9.4) | 43 (12.4) | 16 (10.7) |
| 3–5 years | 126 (11.3) | 51 (8.2) | 67 (19.3) | 8 (5.4) |
| 5–10 years | 284 (25.4) | 137 (22.1) | 123 (35.3) | 24 (16.1) |
| 10–15 years | 183 (16.4) | 123 (19.9) | 33 (9.5) | 27 (18.1) |
| $>$ 15 years | 332 (29.7) | 207 (33.4) | 53 (15.2) | 72 (48.3) |

Abbreviation: N/A: Not applicable.

T-test comparison for sample age distribution and ($\chi^2$) for sex difference analysis in order to assess representativity. Data from paramedics' companies not available for representativity analysis.

institutional nurses mean age (*p = 0.1*), while the mean age of participating physicians was lower than the institutional mean (*p = 0.001*). Gender distribution revealed an overrepresentation of male participants for nurses (*p = 0.03*) but was balanced for physicians (*p = 0.1*).

## Awareness of sepsis

The vast majority (98.5%) of participants were familiar with the word "sepsis" (97.4% of nurses, 100% of physicians and 99.3% of paramedics). Participants were asked to evaluate their knowledge and management skills on sepsis using a 5-category (very good/good/average/fair/poor) Likert scale (Fig 2). Overall, 26.3% of participants graded their knowledge as very good or good (Fig 2A). Similarly, 35.8% graded their management skills as very good or good. An analysis by category of health care professionals revealed similar trends (Fig 2B) although statistically significant differences between professions were noted with physicians self-evaluating best and paramedics self-evaluating worst, whether regarding knowledge or management. We then asked participants to provide answers regarding their perception of sepsis (medical emergency, morbidity/mortality, evaluation, its link to organ dysfunction/propensity to develop under antimicrobial therapy). Participants were cognizant of the severity and the necessity for emergent management of sepsis (87.4 and 95.6%, respectively, strongly agree or agree) (Fig 3). They estimated sepsis and septic shock mortality to be 40% and 50%, respectively. They recognized the association between organ dysfunction and infection for sepsis and that it can arise under antimicrobial therapy (Fig 3). A majority of participants (74.9%, 67.7% and 96.1% respectively) identified age, active cancer, and immunosuppression as risk factors but only half (52.8%) recognized a prior septic event as such.

We next assessed sepsis training. In general, 69.4% of HCPs reported prior training on sepsis. Because our study launched in January 2020, we looked at the years 2017–2019 as the period for training including the 3$^{rd}$ draft of consensus definitions. The majority of participants (73.7%) reported no sepsis-specific training in the last 3 years and 31.6% reported never having attended a sepsis-specific formation (Table 2). Conversely, 26.3% of participants reported training within the last three years. Nurses (82.9%) and paramedics (75.8%) reported more often no training or training more than 3 years before when compared to physicians (56.6%).

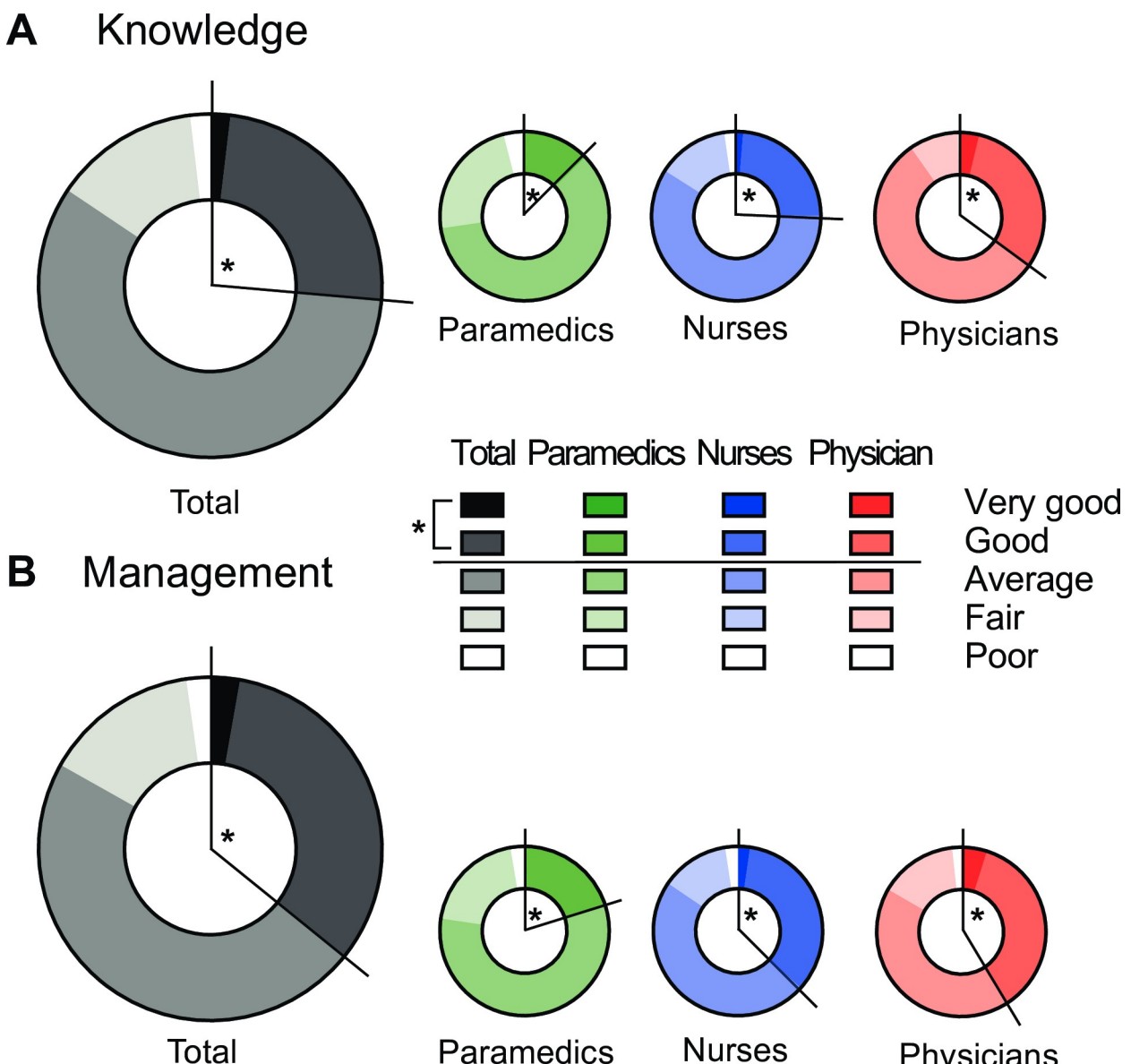

**Fig 2. Assessment of sepsis knowledge and management skills.** Pie chart representation of responses of participants/respondents according to a five-category Likert scale. Assessment of baseline sepsis knowledge and management skills by either the entire study group (A) or by each category of health care professionals (B). Number of participants/respondents: 619 nurses, 358 physicians, and 149 paramedics.

### Knowledge: Definition, detection, diagnosis and management of sepsis

The fraction of participants defining sepsis according to the Sepsis-3 definitions was 12.9% (5.9%, 28.4% and 6.8% of nurses, physicians, and paramedics respectively) (Fig 4A). Nearly half (43.5%) of participants defined sepsis as infection in combination with a systemic inflammatory response syndrome (SIRS) (46.2%, 38.2%, and 49.7% of nurses, physicians, and paramedics respectively) and a quarter (26.9%) as an infection with hemodynamic instability (17.2%, 33.0% and 24.2% of physicians, nurses, and paramedics respectively). A minority of participants defined sepsis as a bacteremia (14.4%) or infection not responding to antimicrobial therapy (2.2%). Knowledge of Sepsis-3 definitions was 40.0% amongst ED physicians,

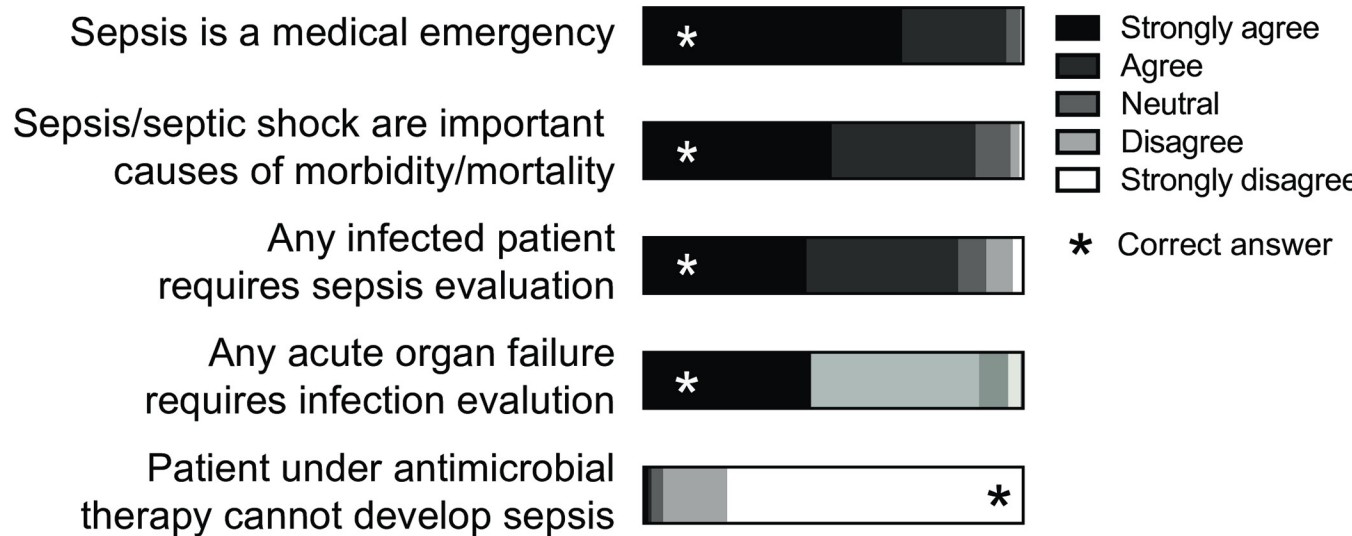

**Fig 3. Sepsis awareness.** Assessment of sepsis characteristics or features (i.e. urgency of care, severity, need for prompt evaluation, and context of appearance) according to a 5-category Likert scale by study participants. Asterisks represent the expected answer.

34.8% amongst ICU physicians, 36.3% among internal medicine ward physicians, 13.3% among surgeons, and 5.6% among psychiatrists. When asked to define by choosing items defining septic shock amongst five components (hemodynamic instability requiring vasopressors despite adequate volume resuscitation/SIRS score > 2 points/ bacteremia / blood lactate > 2 mmol/l / SOFA score > 10 points), 17.0% of physicians defined septic shock according to Sepsis-3 (hemodynamic instability requiring vasopressors despite adequate volume resuscitation and serum lactate of more than 2 mmol/l). Finally, nearly 50% of the physicians associated the qSOFA (Fig 4B) and SOFA (Fig 4C) scores with sepsis. Yet, only 42.1% of physicians reported having computed the SOFA score previously and 17.0% correctly identified the components of the qSOFA score.

Participants were then asked what the recommended timing for intervention was (choice: within 1h/3h/6h/12h/24h), the vast majority of participants (88.5%) chose interventions within one to three hours of sepsis recognition. Fig 5 is a clinical vignette of a patient with suspected sepsis and a qSOFA score of 2 assessing the use of diagnostic tools and management skills shown by the participants according to profession. Nearly all paramedics (90.6%) recognized the need for a rapid transfer to ED (Fig 5A). However, 42.3% considered vital signs monitoring as warranted. The vast majority of nurses recognized the need for immediate medical assessment (93.1%), monitoring of vital signs (82.3%). Most requested blood cultures (70.1%) and half requested drawing blood for laboratory analysis (51.2%) (Fig 5B). Physicians identified vital signs monitoring (92.0%), blood culture draw (96.0%), lactate measurement (89.1%), and

**Table 2. Specific sepsis training among study participants.**

| Timing of last sepsis training | Overall | Nurses | Physicians | Paramedics |
|---|---|---|---|---|
| < *1 year ago* | 109 (9.8) | 30 (4.8) | 70 (20.1) | 9 (6.0) |
| **1–2 years ago** | 97 (8.7) | 29 (4.7) | 54 (15.5) | 14 (9.4) |
| **2–3 years ago** | 87 (7.8) | 47 (7.6) | 27 (7.8) | 13 (8.7) |
| **> 3 years ago** | 470 (42.1) | 241 (38.9) | 152 (43.7) | 77 (51.7) |
| **Never** | 353 (31.6) | 45 (12.9) | 36 (24.2) | 272 (43.9) |

**A**

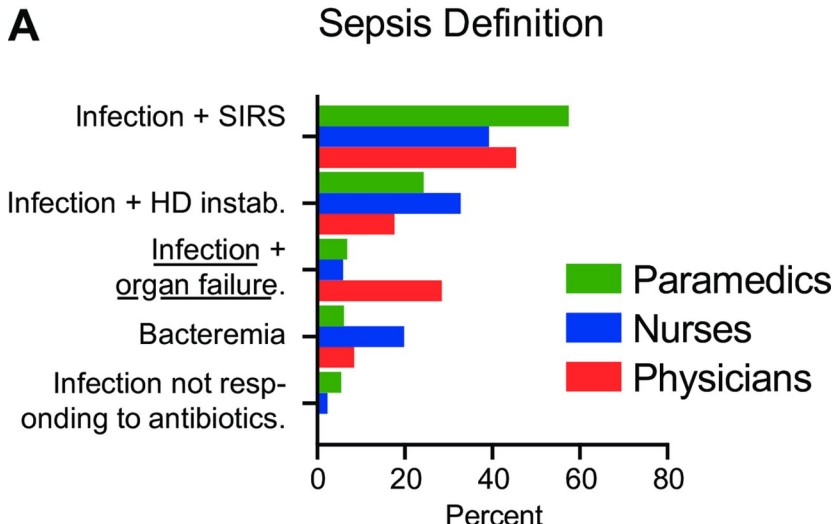

**B**

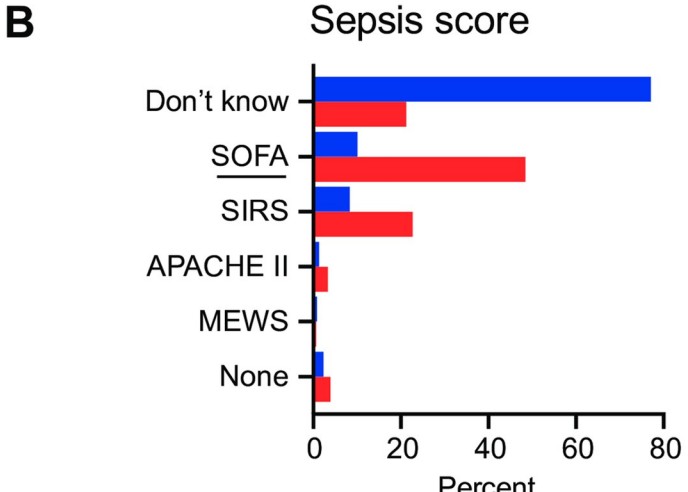

**C**

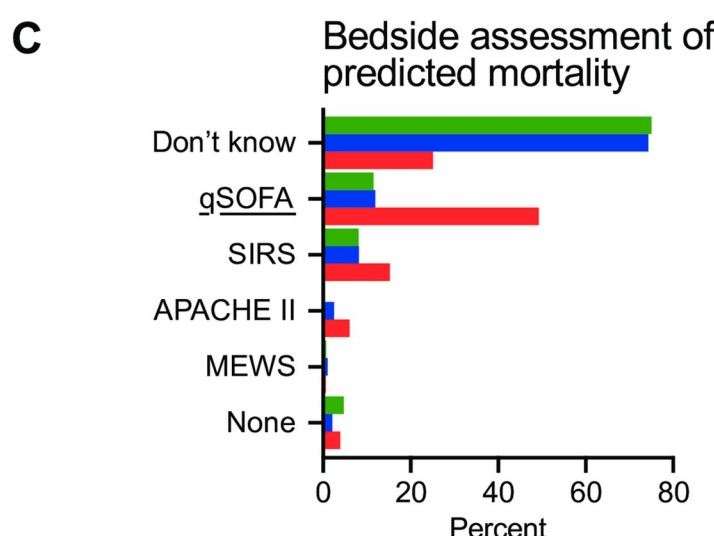

**Fig 4. Sepsis definitions and sepsis scores.** Evaluation of the definition of sepsis (A) and of scores (SOFA, SIRS, qSOFA, MEWS and APACHE II) as a sepsis defining tool (B) or a bedside predictor of sepsis mortality (C). Underlined answers are correct. Abbreviations: MEWS (modified early warning score), APACHE II (acute physiology and chronic health evaluation II).

imaging (77.9%) as critical diagnostic steps (Fig 5C, diagnostic tests). Once sepsis was confirmed (presence of infection plus a SOFA score of 3), the majority of physicians chose to administer broad-spectrum antibiotics (91.7%), to confirm an intravenous access (87.1%) and to start fluid resuscitation (76.1%) as immediate therapeutic interventions (Fig 5C therapeutic interventions).

## Factors associated with sepsis awareness and knowledge

Finally, we looked at associations of participants' characteristics with their sepsis knowledge. In multivariable analyses (Fig 6), the knowledge of SOFA and qSOFA scores' purpose was associated with last sepsis training within the last 3 years, profession experience and self-evaluation of sepsis knowledge for nurses (Fig 6A). For physicians, sepsis training within the last 3 years correlated with knowledge of definitions (Fig 6B). As for nurses, the physicians' knowledge of SOFA and qSOFA scores' purpose was associated with a prior sepsis training within the last 3 years and self-evaluation of sepsis knowledge. Conversely, physician's professional experience correlated inversely with knowledge of the qSOFA score's purpose. Finally, physicians' self-evaluation of sepsis knowledge and recent sepsis training correlated with knowledge of qSOFA score items (Fig 6B). Paramedics did not have factors associated with knowledge of sepsis definitions and only good or excellent self-evaluation correlated with knowledge of qSOFA purpose and its items (Fig 6C).

## Discussion

Our study is a foundational analysis of the sepsis quality of care improvement project at LUH for the strategic development plan of the 2019–2023 period. We identified significant deficiencies in sepsis awareness amongst nurses and physicians of our university tertiary care center and paramedics transporting patients to our hospital. A minority of healthcare professionals in our institution know of Sepsis-3 consensus definitions for sepsis. Similarly, a minority of staff know of SOFA and qSOFA scores. Correspondingly, a minority of paramedics, nurses, and physicians self-evaluated as good or very good for sepsis knowledge and management. Importantly, these findings are associated with a lack of continuing education.

Despite the fact that Sepsis-3 consensus was released four years prior to the survey [1], despite its incorporation into the Lausanne medical school curriculum or in institutional tools such as the LUH's guide for empirical antimicrobial therapy, our results show a lack of uptake of the latest sepsis definition [28, 29]. The lack of specific continuing education accounts primarily for this. Only 18.5% of participants reported having attended sepsis-specific training in the previous three years. Thus, the vast majority of participants have not been exposed to training on the new sepsis definitions and are not familiar with the qSOFA score. This was striking for both paramedics and nurses that are at the front line of sepsis recognition. Nurses spend comparatively more time than physicians at the patient bedside [30] and early recognition of nosocomial sepsis by nurses increases 30-days survival [31].

Similarly, only one-third of physicians know of the current sepsis definition. One-fifth of physicians using the definition of hemodynamic instability in addition to infection, may lead to delays in the recognition of septic patients. Furthermore, the low rate of calculation of a SOFA score by physicians implies that documentation of sepsis in discharge summaries and

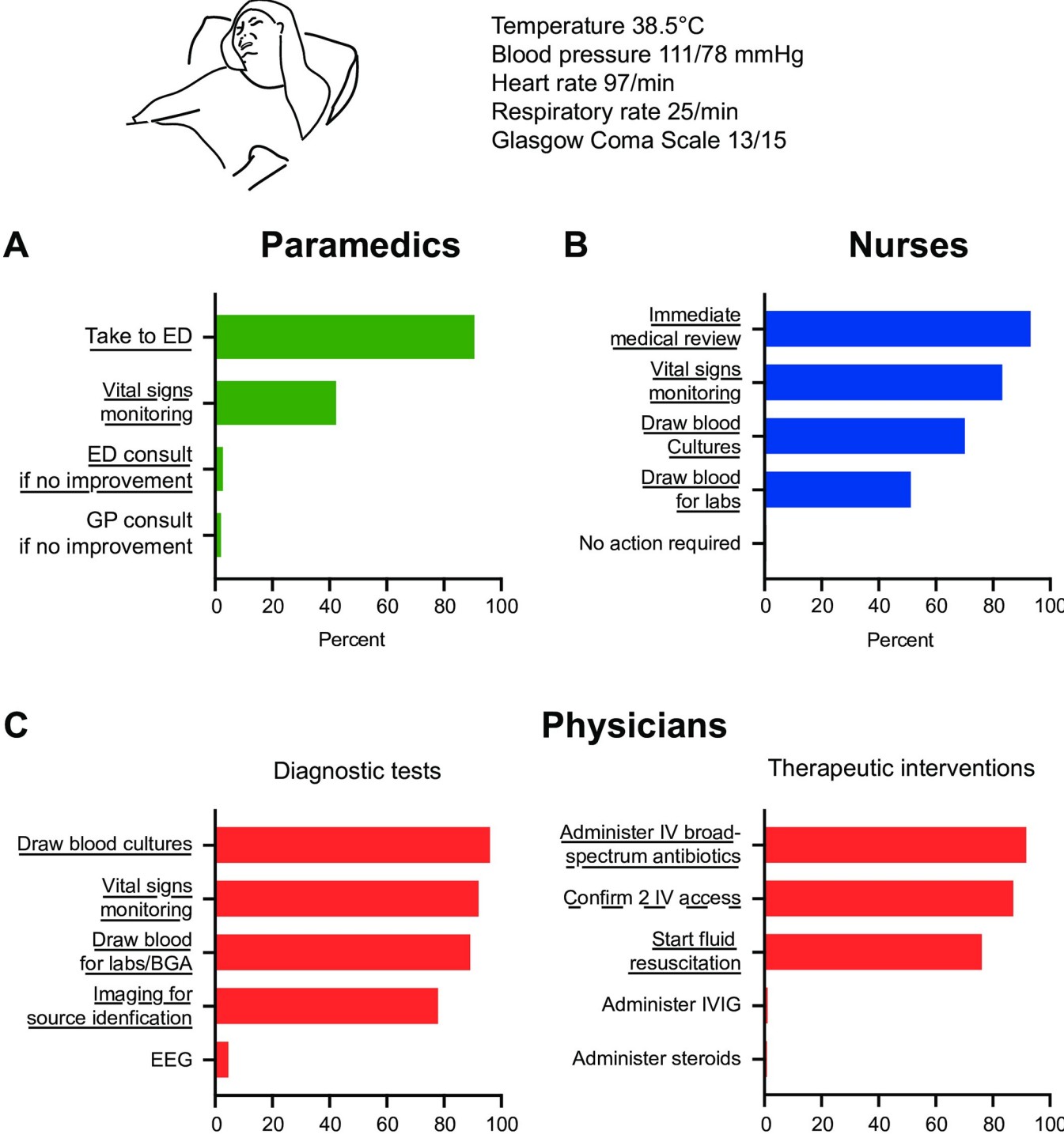

**Fig 5. Clinical vignette.** Management of a patient with suspected sepsis and a qSOFA of 2 (i.e., respiratory rate of 25 per min and a Glasgow Coma Scale score of 13) by paramedics (A), nurses (B), or physicians (C). In panel C, assessment of evaluation tools (step 1) and of management (step 2). Abbreviations: GP (general practitioner), ED (emergency department), EEG (electroencephalogram), IVIG (intravenous immunoglobulins). Underlined answers are expected answers, dotted underlined answers should be considered.

| Outcome | I/C | Variable | OR (IC95%) | OR | p-value |
|---|---|---|---|---|---|
| Nurses | Sepsis definition | 583/36 | age | | 1.052 | 0.013 |
| (N=619) | | | Sepsis training | | 1.817 | 0.171 |
| | | | Sepsis training within the last 3 years | | 2.157 | 0.106 |
| | | | Professional experience | | 0.628 | 0.401 |
| | | | Good/excellent self-evaluation on sepsis knowledge | | 0.785 | 0.571 |
| | Purpose of SOFA score | 557/62 | age | | 0.987 | 0.488 |
| | | | Sepsis training | | 2.392 | 0.043 |
| | | | Sepsis training within the last 3 years | | 3.956 | 0.000 |
| | | | Professional experience | | 2.861 | 0.024 |
| | | | Good/excellent self-evaluation on sepsis knowledge | | 2.490 | 0.003 |
| | Purpose of qSOFA score | 546/73 | age | | 0.983 | 0.316 |
| | | | Sepsis training | | 1.482 | 0.274 |
| | | | Sepsis training within the last 3 years | | 5.084 | 0.000 |
| | | | Professional experience | | 3.147 | 0.007 |
| | | | Good/excellent self-evaluation on sepsis knowledge | | 1.677 | 0.075 |
| | SOFA items propensity | 613/6 | age | | 1.012 | 0.821 |
| | | | Sepsis training | | >10 | 0.996 |
| | | | Sepsis training within the last 3 years | | 0.938 | 0.946 |
| | | | Professional experience | | 0.467 | 0.557 |
| | | | Good/excellent self-evaluation on sepsis knowledge | | >10 | 0.995 |

| Outcome | I/C | Variable | OR (IC95%) | OR | p-value |
|---|---|---|---|---|---|
| Physicians | Sepsis definition | 249/99 | age | | 0.948 | 0.032 |
| (N=348) | | | Sepsis training | | 0.678 | 0.344 |
| | | | Sepsis training within the last 3 years | | 1.839 | 0.041 |
| | | | Professional experience | | 1.186 | 0.592 |
| | | | Good/excellent self-evaluation on sepsis knowledge | | 1.273 | 0.389 |
| | Purpose of SOFA score | 179/169 | age | | 0.951 | 0.020 |
| | | | Sepsis training | | 0.698 | 0.335 |
| | | | Sepsis training within the last 3 years | | 2.617 | 0.000 |
| | | | Professional experience | | 0.669 | 0.186 |
| | | | Good/excellent self-evaluation on sepsis knowledge | | 2.168 | 0.005 |
| | Purpose of qSOFA score | | age | | 0.999 | 0.943 |
| | | | Sepsis training | | 0.989 | 0.975 |
| | | | Sepsis training within the last 3 years | | 2.291 | 0.002 |
| | | | Professional experience | | 0.430 | 0.004 |
| | | | Good/excellent self-evaluation on sepsis knowledge | | 1.648 | 0.062 |
| | qSOFA items proficiency | | age | | 1.038 | 0.118 |
| | | | Sepsis training | | 0.453 | 0.114 |
| | | | Sepsis training within the last 3 years | | 2.388 | 0.026 |
| | | | Professional experience | | 0.645 | 0.262 |
| | | | Good/excellent self-evaluation on sepsis knowledge | | 3.550 | 0.000 |

| Profession | Outcome | I/C | Variable | OR | OR | p-value |
|---|---|---|---|---|---|---|
| Paramedics | Sepsis definition | 139/10 | age | | 0.986 | 0.781 |
| (N=149) | | | Sepsis training | | >10 | 0.993 |
| | | | Sepsis training within the last 3 years | | 2.801 | 0.212 |
| | | | Professional experience | | 2.026 | 0.498 |
| | | | Good/excellent self-evaluation on sepsis knowledge | | 0.986 | 0.988 |
| | Purpose of qSOFA score | 132/17 | age | | 1.060 | 0.128 |
| | | | Sepsis training | | 1.780 | 0.523 |
| | | | Sepsis training within the last 3 years | | 2.508 | 0.219 |
| | | | Professional experience | | 0.472 | 0.407 |
| | | | Good/excellent self-evaluation on sepsis knowledge | | 5.526 | 0.009 |
| | qSOFA items proficiency | 145/4 | age | | 0.919 | 0.371 |
| | | | Sepsis training | | >10 | 0.996 |
| | | | Sepsis training within the last 3 years | | 1.446 | 0.802 |
| | | | Professional experience | | 0.409 | 0.555 |
| | | | Good/excellent self-evaluation on sepsis knowledge | | 21.625 | 0.018 |

**Fig 6. Multivariable analysis.** Notes: Abbreviations: C/I correct/incorrect; OR: Odds ratio; IC 95%: Confidence interval. OR: gray vertical line is 1.0; the heavy black line represents the OR and the light blacklines represent the spread of the confidence interval.

electronic medical records is also compromised. As a consequence, sepsis epidemiology at the institutional level may be severely affected.

These observations support further—and regular—training incorporating Sepsis-3 consensus definitions in our institution as studies support continuous training to improve sepsis awareness amongst participants [32, 33]. Because a minority of participants, whether nurses, paramedics, or physicians rated their knowledge and management skills as good or very good, there is a major opportunity for continuing education.

In considering this study, it is worth mentioning the fact that, with its new draft recommendations dating from after our survey, the Surviving Sepsis Campaign has recommended against using the qSOFA score alone as a single screening or rule-out tool due to low sensitivity [8, 34, 35]. Nevertheless mastering this simple bedside score allows for the rapid identification of adult patients with suspected infection in out-of-hospital, emergency department, or general hospital ward settings that are more likely to have poor outcomes typical of sepsis as a bedside rule-in tool as was elegantly discussed by Mervyn Singer and Manu Shankar-Hari [36]. it remains an important tool for clinicians to master.

To the best of our knowledge, the present study is the first to assess sepsis-3 knowledge with a large sample size survey of multiple professions across all adult departments of a tertiary care center, thus representing all individuals implicated in adult sepsis care. Multiple studies have assessed sepsis awareness [9, 11, 13–19]. However, only three probed Sepsis-3, all of which were limited in scope: Nucera and coworkers assessed Sepsis-3 awareness among nurses and physicians and found similar deficiencies; however, the study was limited to 181 persons and excluded oncology wards. Consistent with our study, they identified major deficiencies in awareness particularly pertaining to scores and definitions. The large sampling in our study, however, enables a better resolution of deficiencies. As an example, the capacity to define sepsis according to Sepsis-3 was significantly better in ICU, ED, and internal medicine than in surgery and psychiatry. Mulders and coworkers assessed a very different setting, interviewing general practitioners, but found similar observations with very low penetrance of SOFA score-based sepsis definitions and qSOFA score-based assessment. Finally, a survey limited to ICU physicians in China revealed limited familiarity with only 16% of 366 physicians using Sepsis-3 consensus definitions [12]. Studies relating to Sepsis-2 definitions had already identified significant deficiencies: Seymour and coworkers found paramedical staff struggling to define sepsis [9]. Abdul Rahman and colleagues identified deficiencies among nurses and physicians in the ED [13]. However, sepsis-specific training is associated with significant improvement in such deficiencies [19].

This study's strengths include the number of participants, the participation rate, the combined assessment of nurses, physicians, and paramedics, and the breadth of departments of adult medicine assessed. Furthermore, the methodology with direct supervision of participants taking the survey ensures high-quality data collection. It also has limitations: The survey was built on perception, knowledge, attitude, and practice of health care professionals towards sepsis based on literature review and focus groups of expert clinicians [37]. It was tested in iterative pilots and revisions among intended respondents. However, we did not perform subsequent reliability (internal consistency, test-retest reliability, or inter-rater reliability) or construct validity assessment through a Crohnbach's alpha test due to the various formats of the questions. Second, it is limited to a single center and results may not be generalizable, although they are consistent with previous studies. Third, we have a slight imbalance towards younger age for participants and male sex for nurses. The exclusion of staff not having daily contact with patients likely accounts in part for the age bias. The propensity of male nurses to take the test is more difficult to explain; it might reflect a more prevalent part-time activity amongst female HCPs compared to male HCPs (average full time equivalent 0.73 vs. 0.82). Fourth, we had significant discrepancies in the various hospital departments. This was strongly influenced by differences in availability (seminars, availability on the ward) of personnel, in part due to the COVID-19 pandemic that broke out shortly after the start of our study.

## Conclusion

Our study reveals significant deficiencies in sepsis awareness at an institutional level, in all professions and departments four years after the introduction of Sepsis-3 consensus definitions.

Their uptake is limited and bedside tools are not mastered. It is associated with a lack of specific training, setting the roadmap for sepsis-education, targeting all professions tailored to their activity. The improved recognition and monitoring among nurses and paramedics and definition implementation among physicians with sustained continuing education is a critical step to our quality of sepsis care improvement program.

## Supporting information

**S1 Checklist. Checklist for Reporting of Survey Studies (CROSS).**
(DOCX)

**S2 Checklist. STROBE statement—checklist of items that should be included in reports of observational studies.**
(DOCX)

**S1 Table. Results of the univariable logistics regression analysis.** Only variables having a significant effect (p-value $\leq$ 0.05) are included.
(DOCX)

**S1 File. Survey translation, nurses version.**
(DOCX)

**S2 File. Survey translation, paramedics version.**
(DOCX)

**S3 File. Survey translation, physicians version.**
(DOCX)

**S4 File. Survey nurses French (original).**
(PDF)

**S5 File. Survey paramedics French (original).**
(PDF)

**S6 File. Survey physicians French (original).**
(PDF)

## Acknowledgments

We thank Ingrid Gilles for the survey design and all the physicians, nurses, and paramedics who participated in the focus groups for the survey validation.

We thank Isabelle Guilleret, Vassili Soumas, and Fady Fares from the LUH Clinical Trial Unit. We thank Nicolas Meylan for his proof-reading and Michael Lobritz for insightful comments on the manuscript and Matthias Cavassini for helping us with the data collection methodology.

## Author Contributions

**Conceptualization:** Rachid Akrour, Isabelle Lehn, Jean-Blaise Wasserfallen, Thierry Calandra, Sylvain Meylan.

**Data curation:** Jean Regina, Tapio Niemi, Santino Pepe, Sylvain Meylan.

**Formal analysis:** Jean Regina, Marie-Annick Le Pogam, Tapio Niemi, Santino Pepe, Sylvain Meylan.

**Investigation:** Jean Regina, Sylvain Meylan.

**Methodology:** Jean Regina, Marie-Annick Le Pogam, Thierry Calandra, Sylvain Meylan.

**Project administration:** Sylvain Meylan.

**Resources:** Thierry Calandra, Sylvain Meylan.

**Software:** Tapio Niemi.

**Supervision:** Marie-Annick Le Pogam, Thierry Calandra, Sylvain Meylan.

**Validation:** Thierry Calandra, Sylvain Meylan.

**Visualization:** Sylvain Meylan.

**Writing – original draft:** Jean Regina, Marie-Annick Le Pogam, Thierry Calandra, Sylvain Meylan.

**Writing – review & editing:** Marie-Annick Le Pogam, Thierry Calandra, Sylvain Meylan.

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
