## [Decision Letter · Decision Letter 0]

17 Jun 2022

PONE-D-22-01684Sepsis awareness at the university hospital level: a survey-based cross-sectional studyPLOS ONE

Dear Dr. Meylan,

Thank you for submitting your manuscript to PLOS ONE. After careful consideration, we feel that it has merit but does not fully meet PLOS ONE’s publication criteria as it currently stands. Therefore, we invite you to submit a revised version of the manuscript that addresses the points raised during the review process. Please respond to the comments question by question and clearly copy the revised writing and point out the page and line number.

We look forward to receiving your revised manuscript.

Kind regards,

Nguyen Tien Huy, Ph.D., M.D.

Academic Editor

PLOS ONE

Journal Requirements:

2. Thank you for stating the following in the Acknowledgments/Funding Section of your manuscript: 

This study was funded in part by the Société Académique Vaudoise. This foundation had no influence on the study design.

Enter: The author(s) received no specific funding for this work.

Reviewers' comments:

Reviewer's Responses to Questions

**Comments to the Author**

1. Is the manuscript technically sound, and do the data support the conclusions?

Reviewer #1: Yes

Reviewer #2: Yes

Reviewer #3: Yes

2. Has the statistical analysis been performed appropriately and rigorously? 

Reviewer #1: Yes

Reviewer #2: Yes

Reviewer #3: I Don't Know

3. Have the authors made all data underlying the findings in their manuscript fully available?

Reviewer #1: Yes

Reviewer #2: Yes

Reviewer #3: Yes

4. Is the manuscript presented in an intelligible fashion and written in standard English?

Reviewer #1: Yes

Reviewer #2: Yes

Reviewer #3: Yes

5. Review Comments to the Author

Reviewer #1: Sepsis awareness of healthcare professionals is an important factor affecting patient survival rates. Therefore, the results obtained from the study will guide the preparation of educational content for the diagnosis and management of sepsis.

There is a need for explanatory information about the sampling method and the questionnaire form in the article.

Suggestions to increase the intelligibility of the article are given below.

Background

Page 3, line 63. The recommendations of the “2021 Sepsis Survival Campaign Guidelines” should be added and these guidelines should be reflected throughout the article.

Methods

Page 4, line 81-83. Was sample calculation used in the research? Used my snowball method? Was consent obtained from the participants in the online online survey? How was the privacy of the participants protected? Information must be added.

Are sepsis care protocols used in any unit of the hospital when specifying the universe characteristics? It must be disclosed.

Measures

Page 4, line 88-89. Were only bibliographies 22 and 23 used when preparing the measurement tool? In this sense, it is understood that quite old and limited resources are used.

Page 4, line 90-96. Which method was used to validate the questionnaire used in the study? Has scope validity index calculation been used for this? What was the expertise of the focus groups on the subject? It should be explained.

Page 4, line 95,96. Nurses are at a key point in delirium diagnosis as well as management. Why weren't nurses' management knowledge and skills measured? It should be explained.

Data collection and recruitement

Page 5. How were the participants prevented from repeating the online survey? It should be explained.

Participants

Page 6, line 138-141. Why was there no stratification in terms of representing the universe among occupational groups? The participation rate of paramedics is very high compared to other groups. The results of the study will be affected by this situation. The number of doctors is misspelled in Figure B (438!).

Definition, detection and diagnosis of sepsis

Page 8. If the sepsis education level of the participants and the experience of encountering a sepsis case were also measured, the power of the study would increase even more. (Can be specified in the limitations of the study)

Page 8, line 178. SOFA > 10 points. What source is this score based on?

Factors associated with sepsis awareness

Page 9, line 201- 208. A table can be given to increase the comprehensibility of the results obtained from "Univariate logistic regression models".

Discussion

The use of q-SOFA was recommended for diagnosing sepsis outside of intensive care units. The final guide discusses its reliability. For this reason, the use of q-SOFA in diagnosing sepsis should be discussed specifically for occupational groups working in intensive care and other fields.

References

The number of current resources can be increased.

Figure 5

Is the case information given here a case for sepsis? Blood pressure does not suggest sepsis! If different information about the case is not given, sepsis may not come to mind first.

Reviewer #2: The authors identified cognitive deficits among physicians, nurses, and paramedics at LUH related to a lack of sepsis-specific training.

Major consideration:

*Abstract: Please provide the research hypothesis, research question, and the method of data analysis.

1. Please provide the research hypothesis or research question in the Introduction

2. How do you calculate sample size?

3. Please specify selection criteria in the Method

4. Please describe any efforts to address potential sources of bias

5. Clearly define all outcomes, exposures, predictors, potential confounders, and effect modifiers.

6. Please note whether the outcome assessor is "blind". Sometimes the person measuring the exposure is the same person conducting the outcome assessment. In this case, the outcome assessor would most likely not be blinded to exposure status because they also took measurements of exposures.

7. Please fill out the checklist and provide missing sections:

7a. STROBE checklist cross-sectional: https://www.strobe-statement.org/checklists/

7b. A Consensus-Based Checklist for Reporting of Survey

Studies (CROSS): https://pubmed.ncbi.nlm.nih.gov/33886027/

8. How did the authors translate the questionnaire?

9. Please clarify your stat analysis of univariate logistic regression or  univariable logistic regression.

Minor consideration

1. Please check your manuscript for grammatical mistakes.

Reviewer #3: This study presents results of a survey for the purpose of quality improvement. The title and abstract are clear. I would suggest that as this article deals only with sepsis knowledge, that the term ‘sepsis awareness’ in the title be changed to ‘sepsis knowledge’

Background is sufficiently developed and the article is appropriate as a QI initiative.

Methods are clearly described and carried out. It is not clear why so many nurses (1810/2463) were screened out – given the criteria of ‘daily contact’ with patients, excluding children. Clarification is needed - about line 113 page 5.

Figure 2 is confusing and needs some clarification. As well, there needs to be some consistency with line 96 (page 4) that indicates that only physicians were asked about management.

As the survey tested knowledge, it would be helpful to ensure that correct answers are provided throughout the Results Section, at appropriate points.

The conclusion could be strengthened by indicating that the survey was about knowledge. The second sentence about lack of mastery of bedside tools goes beyond the data presented. If there is going to be a focus on awareness, in which knowledge is only one component, then there needs to be more information about the context, for example, what “awareness” or information tools are available and where, to all participants.

If the survey is really about awareness, then it may be important to move beyond education to other aspects in the hospital (and in the community for paramedics) that may need attention. Approaches other than education could be mentioned in the Discussion.

However, if this is really about knowledge, with the solution being continuing education, then I would suggest that the focus be ‘tightened’ a bit, to focus more specifically on knowledge, and education.

Overall, this is a timely and interesting QI study, that adds to the international literature on sepsis knowledge in acute care settings.

6. PLOS authors have the option to publish the peer review history of their article (what does this mean?). If published, this will include your full peer review and any attached files.

Reviewer #1: No

Reviewer #2: **Yes: **Van Phu Tran

Reviewer #3: No

---

## [Author Response · Author response to Decision Letter 0]

2 Sep 2022

Editor comments

We formatted the manuscript to adapt PLOS ONE’s style requirements. 

2. Thank you for stating the following in the Acknowledgments/Funding Section of your manuscript: 

This study was funded in part by the Société Académique Vaudoise. This foundation had no influence on the study design.

Enter: The author(s) received no specific funding for this work.

The société académique vaudoise enabled the acquisition of iPads for the survey but did not pay the salary of the MD-Student. We removed the funding statement from the manuscript and changed the funding statement for this sentence: « This study was funded by the Société Académique Vaudoise. This foundation had no influence on the study design. » to “We thank the Soicété Académique vaudoise for their support.”

We did so for ethical reasons. Indeed, some of the subgroups of healthcare workers are so small that people could be identified (e.g. in the psychiatry team, where only 5 physicians participated) through linking sex, age and position. People could then be unmasked. As we intend to uphold our promise to participants that their answers cannot be traced back to them, we would therefore request to maintain our status as such.

We have included in the cover letter a paragraph to address this point.

We removed this sentence from the manuscript.

We moved the ethics statement to the end of the Methods section of the manuscript. 

Reviewer 1

Sepsis awareness of healthcare professionals is an important factor affecting patient survival rates. Therefore, the results obtained from the study will guide the preparation of educational content for the diagnosis and management of sepsis.

We thank reviewer 1 for highlighting the relevance of our observations and their role in continuing education.

There is a need for explanatory information about the sampling method and the questionnaire form in the article.

We weighed 2 options in designing our sampling method: 

- A random sampling by email contacting all HCWs. This approach is limited by several factors. However, we anticipated a recruitment bias (people more versed in sepsis would be more likely to answer). Moreover, unsupervised surveys also jeopardize the quality of data as participants could open a browser and look up the answers. Finally, our experience in our institution is that random sampling in surveys generally reach 10% response rate.

- In contrast an in person approach whereby study coordinators approach HCW directly enabled a supervised data collection (addressing potential data quality issues). It was expected that large samples could be collected. Moreover, collection of demographic data matched to HR statistics enables the evaluation of representativity of such a sampling (described in the result section under “participant”). 

We favored the 2nd approach with random access to HCWs in continuing education seminars. and had aimed initially for 20% participation rate. This was severely challenged by the COVID-19 pandemic. Our demographics analysis helped assess potential bias and found an overall good representativity of the workforce.

To better highlight this the Data collection and recruitment paragraph of the methods section was changed to: 

“Participants were recruited between January 20 and October 10, 2020. We aimed for a convenience sample size of 1,000 persons (approx. 20% of the active HCPs) distributed over all departments (Emergency department (ED), intensive care unit (ICU), Medicine, Paramedic, Psychiatry, or Surgery) and professions (paramedics, nurses and physicians) to reach 20% of LUH staff considered HCPs, being as representative as possible. Pediatrics and neonatology staff (not covered by Sepsis-3 consensus definitions) as well as nurses and physicians not in daily contact with patients (i.e., who were working in research team or in administration) were excluded. We favored a supervised approach rather than a dissemination of the survey to all HCPs by email. Participants answered the online survey under trained interviewer supervision so as maximize data quality and to avoid biased responses (internet queries, discussions between colleagues). Furthermore, to avoid multiple answers by a same HCP, surveys were accessed by QR-code only available at screening; timing of survey completion was registered and email addresses were registered. Thus, participants were screened amongst the medical (n=1664) and nursing staff (n=2463) in daily contact with patients of LUH and amongst paramedics of the Canton of Vaud (n=290) during the screening period. Screening by trained interviewers took place during scheduled patient hand-offs, seminars or group meetings, as permitted by heads of units. Participation was voluntary and anonymous. Participants completed the online survey using tablets or smartphones (participants’ or provided by the investigators).”

Suggestions to increase the intelligibility of the article are given below.

Background 

Page 3, line 63. The recommendations of the “2021 Sepsis Survival Campaign Guidelines” should be added and these guidelines should be reflected throughout the article.

We thank you for this note. It should be cautioned that data collection for the survey was collected a year prior to the release of the surviving sepsis campaign update. While the role for qSOFA has been reframed, this score has not been eliminated. Other elements of the survey remain valid. 

İn order to address the concerns, follow steps have been taken:

we have added the reference of the SSC 2021 draft in the introduction.

The discussion addresses the modifications of the SSC 2021

Lines 345-353

In considering this study, it is worth cautioning that, with its new draft recommendations dating from after our survey, the surviving sepsis campaign has recommended against using the qSOFA score alone as a screening or rule-out tool due to decreased lower sensitivity.8,33,34 Nevertheless mastering this simple bedside score rapidly identify adult patients with suspected infection in out-of-hospital, emergency department, or general hospital ward settings who are more likely to have poor outcomes typical of sepsis as a bedside rule-in tool as was elegantly discussed by Mervyn Singer and Manu Shankar-Hari.35 It remains as such an important tool for clinicians to master.

Methods

Page 4, line 81-83. Was sample calculation used in the research? Used my snowball method? Was consent obtained from the participants in the online online survey? How was the privacy of the participants protected? Information must be added.

No sample calculation as we aimed for maximal coverage. 

Ideally, we would have aimed for 65% of coverage, recognizing however that most survey cover much lower percentage (10-20%). We started our data collection in February 2020. Considering COVID-19 which caused a cessation of continuing education rounds, this was simply not possible. We did however have a minimal convenience sample size of 1000 HCWs representing 20% of the active hospital work force.

Privacy of participants was a primary concern for the investigative team. The survey started with an explanatory introduction text (see appendix added to address this issue). We initially consulted with the local IRB who felt this was quality of care and not human research. Nevertheless, the survey had an introduction stating that this was purely on a voluntary basis and no personal analysis would take place. Indeed, the email address taken solely for feedback purposes was immediately decoupled from answers in our database. In addition, participants were informed of the publication intended at the end of the research project.

Are sepsis care protocols used in any unit of the hospital when specifying the universe characteristics? It must be disclosed.

No sepsis protocol was in use at the time of survey although internal medicine and the Emergency department had guidelines for septic patient management guidelines. The drafting of a sepsis protocol is based on the response we gained from this study. This has been included in the study aim, design and setting.

Line 120-121:

“At the time of the survey, no department had an active sepsis program.”

Measures

Page 4, line 88-89. Were only bibliographies 22 and 23 used when preparing the measurement tool? In this sense, it is understood that quite old and limited resources are used.

These are the primary studies used because they encompassed the scope of our work (institutional, multiple professions). While they are older, these are cornerstone studies for sepsis awareness and deemed determinant in this work. we added 2 other surveys that were also had considered. 

Page 4, line 90-96. Which method was used to validate the questionnaire used in the study? Has scope validity index calculation been used for this? What was the expertise of the focus groups on the subject? It should be explained.

The focus group were constituted of nurses, physicians and paramedics of varying seniority. Their primary task was in assessing whether formulations and relevance of questions were adequate. We did not use validity index calculation as our work was intended to test very basic elements such as demographics, key epidemiology data and definitions.

The following was added in the text line 134-136

“The focus group were constituted of nurses, physicians and paramedics of all seniority levels. Their primary task was in assessing whether formulations and relevance of questions were adequate”

Page 4, line 95,96. Nurses are at a key point in delirium diagnosis as well as management. Why weren't nurses' management knowledge and skills measured? It should be explained.

While we agree that nurses – and paramedics - have key competence in delirium screening, they do not receive a pregraduate training specific to sepsis in Switzerland. For this reason, testing on management was deemed unfair. 

Page 5. How were the participants prevented from repeating the online survey? It should be explained.

We used QR codes not made publicly available to access the survey. Repeating the survey was thus less likely, although a person may have attended multiple rounds of continuing education where the survey was proposed. While we cannot rule out that a person took multiple times the survey, the email address necessary to take the survey would limit such an attempt. Furthermore, since a participant taking the survey is notified upon survey completion, an illicit use of an email address would likely be notified.

Timing of survey was registered to minimize survey outside of our supervision, thus reducing snowball effect described by reviewer 1.

The following text was added (line 161-166):

“We favored a supervised approach rather than a dissemination of the survey to all HCPs by email. Participants answered the online survey under trained interviewer supervision so as maximize data quality and to avoid biased responses (internet queries, discussions between colleagues). Furthermore, to avoid multiple answers by a same HCP, surveys were accessed by QR-code only available at screening; timing of survey completion was registered and email addresses were registered.”

Participants

Page 6, line 138-141. Why was there no stratification in terms of representing the universe among occupational groups? The participation rate of paramedics is very high compared to other groups. The results of the study will be affected by this situation. 

We describe the stratification according to specialty in figure 1B. We hope this is consistent with the demand by reviewer X.

Due to our survey implementation choice, using supervised survey rather than universal distribution to maximize data quality, we experienced a blatant overrepresentation of paramedics. This is, to a large extent, due to the COVID-19 pandemic as our data collection started in 01/2020 but was strained by the lockdown and institutional policies drastically reducing continuing education as of 03/2020. However, because of the breakdown analysis, we hope to somewhat counteract this problem.

The number of doctors is misspelled in Figure B (438!).

Thank you for the observation, this element was corrected in the new version of the figure.

Definition, detection and diagnosis of sepsis

Page 8. If the sepsis education level of the participants and the experience of encountering a sepsis case were also measured, the power of the study would increase even more. (Can be specified in the limitations of the study)

We agree with the reviewer. However, this is very difficult to quantify, particularly in view of definition deficiencies in interviewees.

Page 8, line 178. SOFA > 10 points. What source is this score based on?

We would request to clarify this question. The SOFA score (sepsis-related organ failure assessment score) is the defining score for sepsis. Lactate =/>2 mmol/l despite vasopressor in a patient with sepsis is the definition of septic shock.

Factors associated with sepsis awareness

Page 9, line 201- 208. A table can be given to increase the comprehensibility of the results obtained from "Univariate logistic regression models".

We thank the reviewer for this suggestion, which we have implemented as a supplementary table.

Discussion 

The use of q-SOFA was recommended for diagnosing sepsis outside of intensive care units. The final guide discusses its reliability. For this reason, the use of q-SOFA in diagnosing sepsis should be discussed specifically for occupational groups working in intensive care and other fields.

We would argue that the qSOFA is a bed side tool used to have a first evaluation. As discussed by Mervyn Singer et Manu Shankar-Har, “the Sepsis-3 Task Force's proposal of the quick Sequential (Sepsis-related) Organ Failure Assessment (qSOFA) as a simple bedside assessment to “rapidly identify adult patients with suspected infection in out-of-hospital, emergency department, or general hospital ward settings who are more likely to have poor outcomes typical of sepsis.” (Singer and Shankar-Hari, Annals of Internal Medicine • Vol. 168 No. 4 • 20 February 2018) It is not, as such diagnostic. However, the SSC 2021 has noted that qSOFA due to lower sensitivity should not be used as a screening tool alone. Our survey did not argue that the qSOFA should be used alone as a screening tool for sepsis. 

The following comment was added to the discussion:

“In considering this study, it is worth cautioning that, with its new draft recommendations dating from after our survey, the surviving sepsis campaign has recommended against using the qSOFA score alone as a screening or rule-out tool due to decreased lower sensitivity. Nevertheless, mastering this simple bedside score rapidly identify adult patients with suspected infection in out-of-hospital, emergency department, or general hospital ward settings who are more likely to have poor outcomes typical of sepsis as a bedside rule-in tool as was elegantly discussed by Mervyn Singer and Manu Shankar-Hari.”

References 

The number of current resources can be increased. 

With reviewer suggestions made by the editor and reviewers, we have increased the number of references from 32 to 36

Figure 5

Is the case information given here a case for sepsis? Blood pressure does not suggest sepsis! If different information about the case is not given, sepsis may not come to mind first.

We thank you for this comment but would kindly note that hypotension is not a necessary defining feature of sepsis. Sepsis is defined as an infection with an increase of 2 points in the SOFA score. The SOFA score consisting of 6 axes of deterioration with systolic blood pressure being but one. It is our clinical experience that elderly hypertensive patients can already be septic whilst having “seemingly normal” blood pressures. We hope that this provides clarity regarding the scenario.

Reviewer 2 

The authors identified cognitive deficits among physicians, nurses, and paramedics at LUH related to a lack of sepsis-specific training.

Major consideration:

*Abstract: Please provide the research hypothesis, research question, and the method of data analysis.

We thank the reviewer for this comment. 

Our research question is in the Methods section of the abstract “The survey assessed professionals’ perception, knowledge of sepsis epidemiology, definition, recognition and initial evaluation (nurses and paramedics) and sepsis epidemiology, definition, recognition and management (physicians).“ 

In view of the exploratory nature of the study, we did not have any a-priori hypothesis. We added some details on the data analysis at the end of the Methods section.

We look to the editor for guidance on how to formulate this.

1. Please provide the research hypothesis or research question in the Introduction

The aims of the study are described at the beginning of the method section: “As a part of the effort, this study aims to quantify Sepsis-3 consensus awareness amongst nurses and physicians of various clinical units at LUH and local paramedics and identify potential deficits that should be addressed in continuing education.” 

We added a sentence at the end of the introduction to follow STROBE guidelines. “We aimed to have a representative understanding of sepsis awareness for our tertiary center.”

We look to the editor for guidance as to where the study aim should be placed.

2. How do you calculate sample size?

No sample calculation was designed as we had aimed for maximal coverage. Ideally, we would have aimed for 65% of coverage, recognizing however that most survey cover much lower percentage (10-20%). We started our data collection in February 2020. Considering COVID-19 which caused a cessation of continuing education rounds, this was simply not possible. We did however have a minimal convenience sample size of 1000 HCWs representing 20% of the active hospital work force.

3. Please specify selection criteria in the Method

Selection criteria are described in the “data collection and recruitment” paragraph: 

Participants were recruited between January 20 and October 10, 2020. We aimed for a convenience sample size of 1,000 persons (approx. 20% of the active HCPs) distributed over all departments (Emergency department (ED), intensive care unit (ICU), Medicine, Paramedic, Psychiatry, or Surgery) and professions (paramedics, nurses and physicians) to reach 20% of LUH staff considered HCPs, being as representative as possible. Pediatrics and neonatology staff (not covered by Sepsis-3 consensus definitions) as well as nurses and physicians not in daily contact with patients (i.e., who were working in research team or in administration) were excluded. We favored a supervised approach rather than a dissemination of the survey to all HCPs by email. Participants answered the online survey under trained interviewer supervision so as maximize data quality and to avoid biased responses (internet queries, discussions between colleagues). Furthermore, to avoid multiple answers by a same HCP, surveys were accessed by QR-code only available at screening; timing of survey completion was registered and email addresses were registered. 

Thus, participants were screened amongst the medical (n=1664) and nursing staff (n=2463) in daily contact with patients of LUH and amongst paramedics of the Canton of Vaud (n=290) during the screening period. Screening by trained interviewers took place during scheduled patient hand-offs, seminars or group meetings, as permitted by heads of units. Participation was voluntary and anonymous. Participants completed the online survey using tablets or smartphones (participants’ or provided by the investigators).”

4. Please describe any efforts to address potential sources of bias

We undertook several steps:

- We collected the data in person approach whereby study coordinators approach HCW directly enabled a supervised data collection (addressing potential data quality issues). It was expected that large samples could be collected. Moreover, collection of demographic data matched to HR statistics enables the evaluation of representativity of such a sampling (described in the result section under “participant”). 

- In order to prevent participants from repeating the online survey, we used QR codes not made publicly available to access the survey. Repeating the survey was thus less likely, although a person may have attended multiple rounds of continuing education where the survey was proposed. While we cannot rule out that a person took multiple times the survey, the email address necessary to take the survey would limit fraud and would notify a person of an abuse of their email address. Timing of survey was registered to minimize survey outside of our supervision, thus reducing snowball effect described by reviewer 1.

o The following text was added (line 161-166):

o “We favored a supervised approach rather than a dissemination of the survey to all HCPs by email. Participants answered the online survey under trained interviewer supervision so as maximize data quality and to avoid biased responses (internet queries, discussions between colleagues). Furthermore, to avoid multiple answers by a same HCP, surveys were accessed by QR-code only available at screening; timing of survey completion was registered and email addresses were registered.”

5. Clearly define all outcomes, exposures, predictors, potential confounders, and effect modifiers.

We look to the editor for guidance on this question. 

6. Please note whether the outcome assessor is "blind". Sometimes the person measuring the exposure is the same person conducting the outcome assessment. In this case, the outcome assessor would most likely not be blinded to exposure status because they also took measurements of exposures.

We would ask for clarification. The supervisor generally collected data from groups of 5-20 people. İt is not possible for that person to “see” the answers or link participants to their responses.

7. Please fill out the checklist and provide missing sections:

7a. STROBE checklist cross-sectional: https://www.strobe-statement.org/checklists/

We completed the STROBE checklist and modified the text accordingly.

7b. A Consensus-Based Checklist for Reporting of Survey

Studies (CROSS): https://pubmed.ncbi.nlm.nih.gov/33886027/

We completed the CROSS checklist and modified the text accordingly.

8. How did the authors translate the questionnaire?

The questionnaires were translated by the first author (CEFR B2) and validated by the principal investigator (CEFR C2) and Dr. Casiano Barrera-Groba (Consultant at the Royal Sussex County Hospital, Brighton, UK) validated the translation .

9. Please clarify your stat analysis of univariate logistic regression or univariable logistic regression.

We clarified with the term univariable.

Minor consideration

1. Please check your manuscript for grammatical mistakes.

We have reviewed the manuscript for grammatical mistakes.

Reviewer #3: 

This study presents results of a survey for the purpose of quality improvement. The title and abstract are clear. I would suggest that as this article deals only with sepsis knowledge, that the term ‘sepsis awareness’ in the title be changed to ‘sepsis knowledge’

We thank the reviewer for the overall assessment. We would argue that the evaluation of perceptions, the familiarity with the word “sepsis” encompass more than knowledge, hence our use of the term awareness. This is also based on the reference Seymour et al. J Emerg Med 2012 Jun;42(6):666-77. doi: 10.1016/j.jemermed.2011.06.013. Epub 2011 Nov 8.

Background is sufficiently developed and the article is appropriate as a QI initiative.

We thank the reviewer for this kind assessment.

Methods are clearly described and carried out. It is not clear why so many nurses (1810/2463) were screened out – given the criteria of ‘daily contact’ with patients, excluding children. Clarification is needed - about line 113 page 5.

We thank the reviewer for the evaluation. No HCP was actively screened out. Amongst the work force, we could access only a fraction due to the in person format of data collection. This means that amongst the 2463 nurses with regular contact to patients, we would only access 653. COVID-19 is a serious factor as most continuing education seminars were stopped during our data collection period.

To increase clarity, we added on the flowchart in figure 1a the term “Total” for the first line to clarify the difference with the “screened” line.

Figure 2 is confusing and needs some clarification. 

We have added annotations on each pie chart to add clarity. We hope this helps in clarifying the figure.

As well, there needs to be some consistency with line 96 (page 4) that indicates that only physicians were asked about management.

We thank the reviewer for this interesting comment. In the Swiss healthcare system, paramedics are not allowed to diagnose pathologies and have a limited set of interventions they can take for patients being brought in by ambulance. Similarly, nurses do not have prescribing possibility. Furthermore, in undergraduate nursing education, there is no course on sepsis management. Bearing these considerations in mind, it felt unfair to assess management skills of both groups. However, adapted management measures were asked to test reflexes (resuscitation, iv access) knowing how these groups feel about their management skills identifies actionable.

We modified the text as follows:

The survey was revised using feedback from the groups. Survey of nursing staff and paramedics were more focused on screening and initial evaluation and early management whereas physicians were also tested on diagnosis and management. Responses options included Likert-type scales, binary (e.g. “yes/no”) or multiple choices.

As the survey tested knowledge, it would be helpful to ensure that correct answers are provided throughout the Results Section, at appropriate points.

We use the figures to identify the correct answers (the underlined choice is the correct answer). We have now expanded this to figure 3 (asterisk in the correct answer field. Legends include the explanation of the underline.

The conclusion could be strengthened by indicating that the survey was about knowledge. The second sentence about lack of mastery of bedside tools goes beyond the data presented. If there is going to be a focus on awareness, in which knowledge is only one component, then there needs to be more information about the context, for example, what “awareness” or information tools are available and where, to all participants. If the survey is really about awareness, then it may be important to move beyond education to other aspects in the hospital (and in the community for paramedics) that may need attention. Approaches other than education could be mentioned in the Discussion.

However, if this is really about knowledge, with the solution being continuing education, then I would suggest that the focus be ‘tightened’ a bit, to focus more specifically on knowledge, and education.

We thank the reviewer for these comments. We would argue in favor of awareness in the Seymour et al. publication sense described above. Knowing also does not mean using. This transduction of knowledge into action is a critical point of our quality of care program. The linkert scale (figure 3 are already informative about awareness beyond knowledge).

We would therefore remain with awareness as a broader terme compared to knowledge. 

Overall, this is a timely and interesting QI study, that adds to the international literature on sepsis knowledge in acute care settings.

We thank reviewer 3 for this very kind comment.

---

## [Decision Letter · Decision Letter 1]

7 Oct 2022

PONE-D-22-01684R1Sepsis awareness at the university hospital level: a survey-based cross-sectional studyPLOS ONE

Dear Dr. Meylan,

Thank you for submitting your manuscript to PLOS ONE. After careful consideration, we feel that the authors have not completely addressed the comments. Therefore, we invite you to submit a revised version of the manuscript that addresses the points raised during the review process. A response to reviewers point by point is needed. Please copy and paste the changes in the manuscript to the response file too and indicate the page and line number. The authors need to redo the response and add more information to the manuscript as suggested by reviewers.

We look forward to receiving your revised manuscript.

Kind regards,

Nguyen Tien Huy, Ph.D., M.D.

Academic Editor

PLOS ONE

Reviewers' comments:

Reviewer's Responses to Questions

**Comments to the Author**

1. If the authors have adequately addressed your comments raised in a previous round of review and you feel that this manuscript is now acceptable for publication, you may indicate that here to bypass the “Comments to the Author” section, enter your conflict of interest statement in the “Confidential to Editor” section, and submit your "Accept" recommendation.

Reviewer #1: All comments have been addressed

Reviewer #2: All comments have been addressed

Reviewer #3: (No Response)

2. Is the manuscript technically sound, and do the data support the conclusions?

Reviewer #1: Yes

Reviewer #2: Yes

Reviewer #3: Yes

3. Has the statistical analysis been performed appropriately and rigorously? 

Reviewer #1: Yes

Reviewer #2: Yes

Reviewer #3: I Don't Know

4. Have the authors made all data underlying the findings in their manuscript fully available?

Reviewer #1: Yes

Reviewer #2: Yes

Reviewer #3: Yes

5. Is the manuscript presented in an intelligible fashion and written in standard English?

Reviewer #1: Yes

Reviewer #2: Yes

Reviewer #3: No

6. Review Comments to the Author

Reviewer #1: I thank the authors for their clear and detailed responses and for considering all suggestions.

Best regards

Reviewer #2: Thank you for your great effort. The authors have satisfactorily addressed all of my comments.

Hope I can see your publication soon!

Reviewer #3: Thank you for paying close attention to the recommended changes. In my estimation, most of them have been well addressed. A few places remain unclear and need revision:

• On page 4, line 89, gaps in knowledge among medical and paramedical staff were mentioned and referenced. Nurses should also be added to the list as several of your studies include nurses. An additional study, should you want it, that identifies similar knowledge gaps to yours, is Storozuk et al. (2019). A survey of sepsis knowledge among Canadian emergency department registered nurses. Australasian Emergency Care, 22, 119-125. https://doi.org/10.1016/j.auec.2019.01.007

• On page 5 (line 106), the phrase “active sepsis programme” was used. It is not clear whether this is an education programme or a clinical practice program. Please clarify.

• Method – the word “screened” and the phrase “screened out” do not make sense to me. They do not translate well for an international audience and I do not know what is meant. Please use a different word/phrase.

• Writing – There are still quite a few small but important places throughout the manuscript where copy editing for grammar revisions and English language are needed, primarily for apostrophes, correct verb tense, and preposition use.

Again, overall, this is a timely and interesting QI study, that adds to the international literature.

7. PLOS authors have the option to publish the peer review history of their article (what does this mean?). If published, this will include your full peer review and any attached files.

Reviewer #1: No

Reviewer #2: No

Reviewer #3: **Yes: **Martha L.P. MacLeod

---

## [Author Response · Author response to Decision Letter 1]

25 Oct 2022

Reviewer Responses:

Editor comments

None.

Reviewer 1

Reviewer #1: I thank the authors for their clear and detailed responses and for considering all suggestions.

We thank the reviewer for his evaluation.

Reviewer 2 

Reviewer #2: Thank you for your great effort. The authors have satisfactorily addressed all of my comments.

Hope I can see your publication soon!

We are very thankful for the reviewer’s kind words.

Reviewer #3: 

Reviewer #3: Thank you for paying close attention to the recommended changes. In my estimation, most of them have been well addressed. A few places remain unclear and need revision:

• On page 4, line 89, gaps in knowledge among medical and paramedical staff were mentioned and referenced. Nurses should also be added to the list as several of your studies include nurses. An additional study, should you want it, that identifies similar knowledge gaps to yours, is Storozuk et al. (2019). A survey of sepsis knowledge among Canadian emergency department registered nurses. Australasian Emergency Care, 22, 119-125. https://doi.org/10.1016/j.auec.2019.01.007

We thank the reviewer for this comment. We had unfortunately used the Britannica definition of paramedical staff which includes nurses. We have now added specifically the Word “nursing staff” and added the additional reference.

• On page 5 (line 106), the phrase “active sepsis programme” was used. It is not clear whether this is an education programme or a clinical practice program. Please clarify.

We thank the reviewer for this comment. We have clarified the text:

At the time of the survey, no department had an active education or clinical practice sepsis programme.

• Method – the word “screened” and the phrase “screened out” do not make sense to me. They do not translate well for an international audience and I do not know what is meant. Please use a different word/phrase.

We would ask the reviewer to clarify which part of the text is referred to as we did not find the part of the manuscript in question.

• Writing – There are still quite a few small but important places throughout the manuscript where copy editing for grammar revisions and English language are needed, primarily for apostrophes, correct verb tense, and preposition use.

Again, overall, this is a timely and interesting QI study, that adds to the international literature.

We thank the reviewer fort her comment as well. The text has been reviewed by an English linguistic expert at UNIL (Nicolas Meylan) and multiple changes have been introduced throughout to improve the language. The text has, we hope, gained in clarity and legibility.

---

## [Decision Letter · Decision Letter 2]

31 Oct 2022

PONE-D-22-01684R2Sepsis awareness at the university hospital level: a survey-based cross-sectional studyPLOS ONE

Dear Dr. Meylan,

Thank you for submitting your revised manuscript to PLOS ONE. After careful consideration from our stats expert, we feel that it has merit but does not fully meet PLOS ONE’s publication criteria as it currently stands. Therefore, we invite you to submit a revised version of the manuscript that addresses the points raised during the review process.

We look forward to receiving your revised manuscript.

Kind regards,

Nguyen Tien Huy, Ph.D., M.D.

Academic Editor

PLOS ONE

Reviewers' comments:

Reviewer's Responses to Questions

**Comments to the Author**

1. If the authors have adequately addressed your comments raised in a previous round of review and you feel that this manuscript is now acceptable for publication, you may indicate that here to bypass the “Comments to the Author” section, enter your conflict of interest statement in the “Confidential to Editor” section, and submit your "Accept" recommendation.

Reviewer #4: (No Response)

2. Is the manuscript technically sound, and do the data support the conclusions?

Reviewer #4: No

3. Has the statistical analysis been performed appropriately and rigorously? 

Reviewer #4: No

4. Have the authors made all data underlying the findings in their manuscript fully available?

Reviewer #4: No

5. Is the manuscript presented in an intelligible fashion and written in standard English?

Reviewer #4: Yes

6. Review Comments to the Author

Reviewer #4: The authors have not completely addressed all comments. The authors ignore our previous request of "A response to reviewers point by point is needed. Please copy and paste the changes in the manuscript to the response file too and indicate the page and line number".

1-Target participants should be presented in the title. What are local paramedics?

2-Awareness is completely different from knowledge. There are several questions for knowledge and awareness, how did the author evaluate the level of knowledge and awareness? The authors need to define the outcomes of the study too.

3-Please cite the source of reporting checklist. The authors do not actually describe all information listed in the checklist such as testing of questionnaire, translation methods of questionnaire, providing questionnaire...

4-Please describe selection criteria. Did the authors exclude trainees?

5-Please explain how to calculate the response rate at 91.8%? does it represent the whole population? why did the authors only contact 1216 participants among 4417 potential health care professionals?

6-Univariable regression analysis (NOT univariate) is not good enough to avoid confounders in association analysis. A multivariable regression analysis should be performed. These analyses should be presented in a table.

7-Values of odds ratio should have one more digit in the values.

7. PLOS authors have the option to publish the peer review history of their article (what does this mean?). If published, this will include your full peer review and any attached files.

Reviewer #4: No

---

## [Author Response · Author response to Decision Letter 2]

22 Dec 2022

Reviewer #4: 

The authors have not completely addressed all comments. The authors ignore our previous request of "A response to reviewers point by point is needed. Please copy and paste the changes in the manuscript to the response file too and indicate the page and line number".

1-Target participants should be presented in the title. What are local paramedics?

The title of the manuscript was modified as requested by the reviewer and now reads “Sepsis awareness and knowledge amongst nurses, physicians and paramedics of a tertiary care center in Switzerland: a survey-based cross-sectional study”. Local paramedics are emergency medical services personnel providing emergency pre-hospital medical care on patient’s site and during transportation to hospital by ambulance. This has been clarified on line 35 page 94, line 94 and 97 page 5, by replacing the text “..local paramedics..” with “..paramedics transporting patients to our hospital”.

2-Awareness is completely different from knowledge. There are several questions for knowledge and awareness, how did the author evaluate the level of knowledge and awareness? The authors need to define the outcomes of the study too.

We thank the reviewer for this comment. We evaluated awareness based on the perceptions of the participant familiarity with the word sepsis. We assessed knowledge of sepsis based on questions focusing on the participants’ information/proficiency/skills on the definition, epidemiology, scores and management of sepsis. The words “awareness” and “knowledge” are integrated into the title of the article. This was also specified in the abstract “Measured outcomes included professionals’ demographics (age, profession, seniority, unit of activity) and quantification of prior sepsis education, awareness and knowledge of sepsis assessed based on the information/proficiency/skills on the epidemiology, scores, definition and management of sepsis”. Likewise, we modified the text in the methods (line 115-119 page 6) that reads “The final survey contained questions on participants’ demographic characteristics (5/7/6 questions for nurses/paramedics/physicians), awareness was characterized by questions on sepsis perception and data on continuing education (3/3/3 questions) and self-evaluation of sepsis knowledge and clinical management (2/2/2 questions); the participants’ knowledge was characterized by questions on the definition, assessment scores, the epidemiology (11/12/14 questions) and the management of sepsis (4/4/5 questions)”. Accordingly, the results section is now partitioned into participants, awareness and knowledge.

3-Please cite the source of reporting checklist. The authors do not actually describe all information listed in the checklist such as testing of questionnaire, translation methods of questionnaire, providing questionnaire...

In revision, we provide the reference of the CROSS checklist used for survey reporting (Sharma A et al. A Consensus-Based Checklist for Reporting of Survey Studies (CROSS). J Gen Intern Med. 2021 Oct;36(10):3179-3187. doi: 10.1007/s11606-021-06737-1.) The questionnaire was built de novo by the authors from references 9, 14, 24, and 25 (lines 103-104 page 5 of the manuscript). It was tested and revised through focus groups conducted with each professional category involved (lines 106-110 page 5 of the manuscript). The questionnaire and the answers to the questionnaire were written in French. In line 121-122 of the manuscript, we indicated that the questionnaire was translated into English by one of the authors and revised by another for publication. The survey was translated internally (JR, SM) but was reviewed externally by colleagues from the Brighton and Sussex University Hospital’s Intensive Care Unit. The English version of the survey is in the supplementary material. We have now added the survey in French.

4-Please describe selection criteria. Did the authors exclude trainees?

We have clarified selection criteria as specified on line 124-133, page 6 ” …including registered nurses, and physicians, including medical residents and fellows having graduated from medical school and who were in training for board certification in a medical specialty and attendings, issued from all departments (Emergency department (ED), intensive care unit (ICU), Medicine, Paramedic, Psychiatry, and Surgery) and professions (paramedics, nurses and physicians) in order to achieve maximum representativity of LUH staff considered as HCPs. Pediatrics and neonatology staff (not covered by Sepsis-3 consensus definitions) as well as nurses and physicians not in daily contact with patients (i.e., those working in research teams or in administration) were excluded. For paramedics, we included those transporting patients to LUH. Undergraduate trainees were excluded.”

5-Please explain how to calculate the response rate at 91.8%? does it represent the whole population? why did the authors only contact 1216 participants among 4417 potential health care professionals?

We favored an in person survey sampling to gain higher quality in participant response (e.g. taking the survey online with basic questions had a risk of participants looking up the definition online). To do so, we accessed HCPs mostly during continuing education rounds on their worksite. This meant that we actually got to meet with 1216 HCP out of the total 4417 HCP of LUH. Please bear in mind that the study, while designed in 2019, was performed during the COVID-19 pandemic when continuing education was severely impacted by social distancing measures. Due to strenuous work conditions, we only had specific time points for interviewing healthcare professionals. The participation rate thus represents the fraction of HCPs who were asked to participate. This is explained on lines 133-143, page 7 and in figure 1.

6-Univariable regression analysis (NOT univariate) is not good enough to avoid confounders in association analysis. A multivariable regression analysis should be performed. These analyses should be presented in a table.

We thank the reviewer for this important point. In our revision, we have replaced the univariable analysis with a multivariable analysis now provided in the abstract (page 3 and lines 51-55) and results section (i.e. on page 12 and lines 240-250 and in Table 3). We proceeded with multivariable regression analyses for each healthcare worker stratum.

7-Values of odds ratio should have one more digit in the values.

Odds ratio values were changed throughout the manuscript as requested by the reviewer.

---

## [Editor Report · Decision Letter 3]

18 Apr 2023

Sepsis awareness and knowledge amongst nurses, physicians and paramedics of a tertiary care center in Switzerland: a survey-based cross-sectional study

PONE-D-22-01684R3

Dear Dr. Meylan,

We’re pleased to inform you that your manuscript has been judged scientifically suitable for publication and will be formally accepted for publication once it meets all outstanding technical requirements.

Kind regards,

Luis Antonio Gorordo-Delsol, MD

Academic Editor

PLOS ONE
---

## [Editor Report · Acceptance letter]

24 Apr 2023

PONE-D-22-01684R3 

Sepsis awareness and knowledge amongst nurses, physicians and paramedics of a tertiary care center in Switzerland: a survey-based cross-sectional study 

Dear Dr. Meylan:

I'm pleased to inform you that your manuscript has been deemed suitable for publication in PLOS ONE. Congratulations! Your manuscript is now with our production department. 

Kind regards, 

on behalf of

Dr. Luis Antonio Gorordo-Delsol 

Academic Editor

PLOS ONE